# Time Spent on Daily Activities and Its Association with Life Satisfaction among Czech Adolescents from 1992 to 2019

**DOI:** 10.3390/ijerph19159422

**Published:** 2022-08-01

**Authors:** Lucia Kvasková, Karel Rečka, Stanislav Ježek, Petr Macek

**Affiliations:** Institute for Research of Children, Youth and Family, Faculty of Social Studies, Masaryk University, Joštova 10, 602 00 Brno, Czech Republic; reckak@mail.muni.cz (K.R.); jezek@fss.muni.cz (S.J.); macek@fss.muni.cz (P.M.)

**Keywords:** daily activities, leisure, life satisfaction, sleep duration, online gaming, sports, school commuting, time spent at school, birth cohorts, adolescents

## Abstract

Life satisfaction, an important precursor of adolescents’ well-being, is linked to daily activities. Substantial changes have been noted in adolescents’ daily activities over the years, raising the question of possible consequences for life satisfaction. This study aimed to explore changes in adolescents’ life satisfaction and their time spent on daily activities (sleeping, engaging in sports, online gaming, school commuting, time spent at school) and further investigate the associations between adolescents’ life satisfaction and these daily activities. The sample comprised 2715 adolescents from birth cohorts surveyed at four time points between 1992 and 2019. Participants were administered the Daily Activities Inventory and the Berne Questionnaire on Adolescents’ Well-Being. Robust ANOVA with post hoc tests and spline regression were employed. We found cohort differences in sleep duration (8.6 h a day on average in 1992 and 7.5 in 2019). Sleep duration of 8 h and 1 h of sports activities had a beneficial effect on life satisfaction, while more than 1 h of online gaming had a negative impact. Neither school attendance nor commuting to school was associated with adolescents’ life satisfaction. The results of this study can inform public-health policies to promote sleeping and sports habits in adolescence.

## 1. Introduction

Life satisfaction, defined as the cognitive dimension of subjective well-being [1,2], is linked to adolescents’ physical and mental health. Low life satisfaction in adolescence has been associated with suicide ideation [3], increased psychopathology [4], obesity [5], school disengagement [6], substance abuse [7], and other risk behaviors [8] that can negatively influence adolescents’ important life outcomes [9,10,11]. Still, research on adolescents’ time use and their psychological well-being over time is scant in Europe [12,13].

Previous research has shown that time spent on daily activities plays an important role in adolescents’ life satisfaction [14,15,16,17]. Moreover, recent studies have suggested that the amount of time adolescents spend on daily activities has changed in the last decades [17,18,19]. Very little is known about how these changes are linked to adolescents’ life satisfaction. To address this research gap, we aimed to examine changes in time use (i.e., selected daily activities) by four different birth cohorts from 1992 to 2019 and the strengths of associations between various daily activities and adolescents’ life satisfaction, while controlling for age, gender, and socioeconomic status (SES) as confounding factors of time use and life satisfaction [20,21,22,23].

Since essential determinants of health and well-being are embedded in daily behaviors [24,25], studies on daily activities can significantly contribute to understanding adolescents’ life satisfaction [14,15,16,17]. However, culture, which changes due to economic, technological, and political trends, significantly impacts adolescents’ time use [26]. That has been especially true for digital technologies and their integration into daily life [17,27].

Young people’s involvement in online gaming (i.e., Internet gaming) is increasing rapidly, resulting in many problematic Internet users in recent years, worldwide and in the Czech Republic [28,29]. Internet gaming has received considerable attention [30], primarily due to its potential for excessive use that may result in Internet gaming disorder, a new diagnosis recently included in the latest version of the Diagnostic and Statistical Manual of Mental Disorders (DSM-5) [31] and the International Classification of Diseases (ICD-11) [32]. However, the link between the time spent on Internet gaming and poorer well-being is unclear. While some studies have shown that online gaming is negatively associated with well-being [17,30], another study [33] demonstrated that light users (<1 h a day) of digital media (smartphones, computers, social media, gaming, Internet) reported substantially higher psychological well-being than heavy users (5+ hours a day). These results suggest that time spent on online gaming might be crucial to psychological well-being.

Previous cohort studies have shown an increase in adolescents’ psychological well-being through the 2000s [34,35]. Its subsequent drop between 2012 and 2016 has been linked to adolescents spending their time differently, specifically the amount of screen time [17]. Heavy levels of time may be spent on screen-based activities at the expense of the time otherwise dedicated to sports and other physical activities [36]. Indeed, a recent study of Czech adolescents [14] identified sports as one of the strongest predictors of mental well-being in adolescence during the lockdown in spring 2020, while time spent with electronic media (for non-school purposes) was a negative predictor. Similarly, some studies have reported thresholds of engagement in sports activities to maintain a beneficial effect. For instance, an analysis of a large representative sample of European adolescents showed that low to moderate frequency of physical activity was positively correlated with well-being and negatively correlated with anxiety and depressive symptoms [37]. However, girls who reported being physically active for at least 60 min every day in the last two weeks had reduced levels of well-being [37]. Consequently, some public-health researchers have called for studying sports benefits, to set the recommended amount of sports activity for adolescents [38].

Engaging in new media screen time has also been linked to decreased sleep duration [27]. While sleep length has been identified as a significant contributor to various well-being indicators, such as life satisfaction [39,40], some studies, primarily those conducted in the United States, have shown a decrease in sleep duration among U.S. adolescents over the last decades [18,19,27]. Adolescents in 2015 were 16%–17% more likely to report sleeping fewer than 7 h a night on most nights, compared to 2009 [27]. However, the American Academy of Sleep Medicine recommends 8 to 10 h of sleep per 24 h for 13–17-year-old adolescents [41]. In the Czech Republic, 34% of 13-year-old adolescents and 54.5% of 14-year-old adolescents meet sleep recommendations on school days [42]. A sample of Czech adolescents showed some associations between well-being and sleep [43]. Furthermore, longer total sleep duration in adolescence was associated with greater life satisfaction, after adjusting for personal and parental factors and family environment [39]. Several studies have aimed to determine the number of hours of sleep individuals need to maintain their well-being. Specifically, it has been shown that sleep duration of fewer than 8 h or more than 9 h on school nights was significantly associated with depressive symptoms compared to a sleep duration of 8 h [16]. Sleep duration of fewer than 7 h has been associated with poorer self-rated health and increased odds of feeling depressed [40]. Thus, there are risks from both short and long sleep durations.

Time spent in non-leisure activities, such as time spent in school and commuting to school, may also affect life satisfaction. Although full-time compulsory education ends at 16 in all U.S. states and approximately half of European countries [44], very little is known about adolescents’ time in school and how this non-leisure time relates to their life satisfaction. One previous study [45] proposed a link between higher school time and lower health-related quality of life in adolescent girls. Although the time spent in school increased by about two hours per week from 1981 to 1997 [46], recent data and research on school time and life satisfaction are lacking.

The relationship between commuting and life satisfaction has gained research attention over the last decades [47]. Evidence suggests that a longer commute time in adults is associated with lower levels of both life satisfaction and happiness, especially when the commute times are extreme (≥1 h per day) [48]. Although the mean distance traveled to school has increased [49], making commuting to school a significant daily activity, few studies have explored the connection between commuting to school and adolescents’ life satisfaction. One recent study [15] showed that commuting time is negatively related to life satisfaction for male and female high school students in the Czech Republic. However, no such relationship has been found in Italian and Slovenian adolescents.

Age, gender, and socioeconomic status (SES) must be considered when examining time use and life satisfaction. Participation in organized sports has been reported to decline during adolescence [20], while engaging in screen time is rising [33]. Concerning gender differences, boys spend more time on physical activities and online gaming [20,21]. Regarding SES, time use of less-affluent families is associated with lower physical activity, higher screen time, and less sleep [22,23].

Researchers have commonly used three categories of daily activities: personal maintenance (e.g., sleeping, personal care, and eating), leisure, and work or non-leisure activities (e.g., commuting to school, schoolwork, or housework) [26,50]. To address the impact of cultural changes more closely, this study focused on specific individual daily activities among those previously robustly connected to life satisfaction in adolescence: online gaming, sports, sleeping, time spent at school, and commuting to school.

## 2. Materials and Methods

### 2.1. Sample and Procedure

Data were obtained from the Euronet Pilot Study [12], a survey run in 1992, and three replications of the survey in 2001, 2011, and 2019. Quota sampling was used to represent the population of grade 8 (around age 14) and grade 10 (around age 16) students in the Czech Republic attending lower and higher secondary schools. Whole classrooms were sampled. Participants in the 1992 and 2001 cohorts completed a pen-and-paper questionnaire, and participants in 2011 completed an online survey. In 2019, schools chose the mode of survey administration (78% of participants completed the survey offline). The first three samples were small (*n* = 257, 310, and 371 in 1992, 2001, and 2011, respectively), while the 2019 sample was large with 3206 participants. The 2019 survey used the 3-form planned missingness design [51], with about half of the measures included in all three survey forms and the remaining measures included in two out of the three survey forms. The forms were assigned randomly (*n* = 1064, 1044, and 1098), and items measuring daily activities were administered to 2162 participants in the 2019 survey.

The proportion of missing data increased over time, in line with the generally observed online data collections (2%, 1%, 18%, and 16%, respectively). The effective sample consisted of 2715 adolescents from four cohorts: 1992 (9%, *n* = 253), 2001 (11%, *n* = 305), 2011 (12%, *n* = 338), and 2019 (67%, *n* = 1819). There were slightly more girls (55%) than boys. Most adolescents (71%) were between 16 and 18 years old.

As expected, most adolescents (67%) stated that their parents earn about the same as other parents. Table 1 summarizes the characteristics of the sample in general and within cohorts.

### 2.2. Measures

#### 2.2.1. Daily Activities

Self-report data on adolescents’ daily activities were collected using the Daily Activities Inventory from Euronet Pilot Study Questionnaire [52]. The participants described daily activities conducted during the working day (from midnight to midnight) preceding the day of data collection (no Sundays or Saturdays). They were given a list of 15 categories of activities and indicated time intervals during which they were involved/engaged in each category. The categories included sleeping, dressing, body care, eating, school commuting, time spent at school, doing homework, playing a musical instrument, reading but not school assignments, shopping, engaging in active sports, working for family/house/mother etc., dating, hanging around with friends, jobbing, and watching TV and listening to the radio (while doing nothing else). A blank space was provided for the participants to add activities that were not listed. In 2019, online gaming, volunteering, social media use, and online video-watching categories were added (e.g., YouTube). Pen-and-paper and online versions differed slightly in the format of the items, with the paper version allowing the participants to select from 15 min time slots vs. the online version asking the participants to enter specific time ranges during which they conducted each activity.

#### 2.2.2. Life Satisfaction

Items assessing adolescents’ life satisfaction were selected from the Berne Questionnaire on Adolescents’ Well-Being (BSW-Y) [53], a 13-item measure containing two first-order scales of subjective well-being/life satisfaction and self-esteem. In the present study, we employed 8 items for the life-satisfaction scale. Response options ranged from 1 = “entirely untrue” to 4 = “entirely true”. The total life satisfaction score was computed as a mean of all 8 items; thus, the minimum and maximum attainable scores were 1 and 4, respectively, with higher scores indicating higher life satisfaction. Using a Robust Maximum Likelihood (MLR) estimator, a one-factor model of the scale had an acceptable fit to the data, χ^2^(28, N = 2506) = 193.96 (scaling factor = 1.18), *p* < 0.001, CFI = 0.940, RMSEA = 0.059 (90% CI [0.052, 0.066]), and SRMR = 0.036. The estimated reliability was also acceptable (Cronbach’s α = 0.72, McDonald’s Ω = 0.73).

### 2.3. Statistical Analysis

We analyzed the data in R, a free software environment for statistical computing and graphics [54], using the tidyverse package [55] to manipulate and visualize the data, the WRS package [56] to perform robust ANOVA with post hoc tests, and the lspline package [57] to estimate linear regression with splines. The first objective of the analysis was to compare the four cohorts in terms of the average time spent sleeping, commuting to school, in school, and playing sports and the average life satisfaction. To reduce the influence of outliers that were not likely to reflect a typical day and to deal with heteroskedasticity, we used Welch’s ANOVA [58] and Dunnett’s T3 post hoc tests [58] to compare 5% trimmed means. The second objective of the analysis was to capture the effects of the time spent sleeping, commuting to school, in school, playing sports, and playing online games on life satisfaction, controlling for gender, age, and self-reported parents’ income (i.e., SES). We expected these effects to be nonlinear. Therefore, we used regression with splines, a flexible approach to capturing non-linear relationships. It is a piece-wise regression, constrained to avoid discontinuities between adjacent segments or splines. Knots, which define where one spline ends and another one starts, are pre-specified values of a predictor. Although splines are frequently fitted as third-degree polynomials, we used linear splines to aid interpretation and achieve more parsimonious models.

## 3. Results

The duration of online gaming ranged from 0 to 11 h (Mtr = 0.52, SDw = 1.27). However, 65% of the respondents reported that they did not play online games at all during the day. The distribution was positively skewed (b1 = 2.88) and leptokurtic (K = 9.15). This variable was measured only in the last (2019) cohort, with eight missing values (0.4% out of *n* = 1819 for the 2019 cohort).

The duration of sports activities ranged from 0 to 5.75 h (Mtr = 0.62, SDw = 0.94), but 54% of the respondents did not report any sports activity during the day. Therefore, the distribution was positively skewed (b1 = 1.52) and leptokurtic (K = 2.15). There were 14 (0.5%) missing values.

The sleep duration ranged from 0 to 12.5 h (Mtr = 7.71, SDw = 1.18), although most (90%) adolescents reported sleeping 5.25 to 9.25 h. The distribution was slightly negatively skewed (b1 = –1.19) and leptokurtic (K = 3.97). There were only five missing values.

Time spent in school ranged from 30 min to 11.5 h (Mtr = 6.03, SDw = 0.97), with 90% of adolescents reporting that they spent 4.25 to 7.5 h at school. The distribution was approximately normal (b1 = 0.44, K = 2.82). There were 31 (1.1%) missing values.

The duration of school commuting ranged from 2 min to 5 h (Mtr = 0.95, SDw = 0.64), but 90% of the adolescents reported that commuting from home to school and from school to home took them 15 min to 2 h. The distribution was positively skewed (b1 = 1.57) and leptokurtic (K = 3.15). There were 310 (11%) missing values. The higher number of missing values, compared to other variables, was due to many respondents misunderstanding the question and reporting the times when they arrived at and left school (instead of the times they left home and arrived at school and the times they left school and returned home).

Life satisfaction scores ranged from 1.0 to 4.0 (Mtr = 2.98, SDw = 0.41), with the middle 90% of the scores falling between 2.12 and 3.50. The distribution was slightly negatively skewed (b1 = −0.64, K = 0.54). There were 31 (1.1%) missing values.

### 3.1. Changes in Daily Activities and Life Satisfaction by Birth Cohorts from 1992 to 2019

First, we compared the cohorts’ sleep duration; the amount of time spent in sports activities, school, and commuting to school; and their life satisfaction. Figure 1 shows the distribution of variables by cohort with boxplots, and Table 2 shows descriptive statistics and ANOVA results together with post hoc tests. The differences between the cohorts were all significant (*p* < 0.05), except for school commuting. However, in terms of effect sizes, differences in sports and life satisfaction were negligible (η^2^ around 0.01). A decrease in sports duration was observed in the 2001 cohort (Mtr = 0.71 h, 95% CI [0.57, 0.85]) compared to the 1992 cohort (Mtr = 0.44 h, 95% CI [0.34, 0.55]; ΔMtr = 0.27 h, 95% CI [0.03, 0.50]), but, subsequently, it increased gradually, so that the 2019 cohort (Mtr = 0.66 h, 95% CI [0.61, 0.71]) differed from the 1992 cohort only marginally (ΔMtr = 0.05 h, 95% CI [−0.15, 0.25])). The differences in sleep duration were the strongest (η^2^ = 0.16). As shown in Figure 1, there was a clear decreasing trend in the average sleep duration between the cohorts. For example, the 1992 cohort reported sleeping on average 8.6 h a day (95% CI [8.48, 8.72]), but the 2019 cohort reported sleeping only about 7.5 h a day (95% CI [7.40, 7.52]), a difference of more than one hour (ΔMtr = 1.14 h, 95% CI [0.96, 1.32]). The differences in school attendance duration were small to medium (η^2^ = 0.05). The lowest average duration of school attendance was noted in the 2001 cohort (Mtr = 5.81 h, 95% [5.70, 5.93]) and the 2011 cohort (Mtr = 5.82 h, 95% [5.63, 6.01]), and the highest duration was observed in the 2019 cohort (Mtr = 6.08 h, 95% CI [6.04, 6.13]). We also tested and confirmed that the effects of the cohort on sleep, school travel, school attendance, sports, and life satisfaction remain significant, even after controlling for the effects of gender, age, and parents’ income.

### 3.2. Associations between Daily Activities and Life Satisfaction

We then predicted life satisfaction based on cohort, gender, age, and parents’ income, using the 2019 cohort, male gender, age under 16 years, and earnings “about the same as other parents” as reference categories for indicator variables. The overall model was significant, F(7, 2606) = 16.88, *p* < 0.001, and explained about 5% of the variance in life satisfaction, R2 = 0.043, adj. R2 = 0.041. The cohort effect on life satisfaction was small but significant. The 1992 cohort (B = −0.07, 95% CI [−0.14, −0.02], *p* = 0.008) and the 2011 cohort (B = −0.072, 95% CI [−0.12, −0.02], *p* = 0.006), but not the 2001 cohort (B = −0.02, 95% CI [−0.08, 0.03], *p* = 0.425), reported significantly lower life satisfaction than the 2019 cohort, when controlling for the effect of other variables. Furthermore, the effect of gender was significant, with girls indicating slightly lower life satisfaction, (B = −0.07, 95% CI [−0.11, −0.04], *p* < 0.001). The effect of parents’ income was also significant. Adolescents who perceived their parents to earn less compared to other parents showed lower life satisfaction (B = −0.18, 95% CI [−0.23, −0.13], *p* < 0.001), while adolescents who perceived their parents to earn more than other parents showed higher life satisfaction (B = 0.09, 95% CI [0.04, 0.13], *p* < 0.001). However, the effect of age was negligible (B = 0.02, 95% CI [−0.01, 0.06], *p* = 0.213). The intercept (representing the predicted value for a male adolescent, younger than 16 years, from the 2019 cohort, whose parents earn about the same as other parents) of the model was B = 3.02, 95% CI [2.97, 3.06]).

Finally, we saved the residuals from the above model. We investigated whether the duration of online gaming, sports activity, sleep, school travel, and school attendance (modeled with splines regression) explain additional variance above and beyond the effects of the cohort, gender, age, and parents’ income. The first and last knots were always set at the maximum and minimum observed values; all other knots were specified manually at substantively relevant values.

First, we tested the effect of playing online games with knots at 0, 1, and 11 h. Since the duration of playing online games was measured only in the 2019 cohort, we did not control for the cohort effects because the cohort was a constant for this subsample. Adding the effect of online games significantly increased the explained variance, ΔR2 = 0.008, F(2, 1718) = 6.76, *p* = 0.001. For 0 to 1 h of gaming, the slope was positive but not significant (B = 0.02, 95% CI [−0.04, 0.08], *p* = 0.589); however, for more than 1 h of gaming, the slope became negative and significant (B = −0.03, 95% CI [−0.05, −0.01], *p* < 0.001). Thus, playing online games appears to have a negative effect on life satisfaction, but only after crossing certain thresholds of about 1 h. Figure 2 visualizes all regression splines, illustrating the effects in an understandable form.

Second, we tested the effect of playing sports (or doing another physical exercise) with knots at 0, 1, 2, and 5.8 h. Adding this effect led to a small but significant increase in the explained variance, ΔR2 = 0.006, F(3, 2594) = 5.35, *p* = 0.001. For a duration of 0 to 1 h, the regression slope was positive (B = 0.06, 95% CI [0.01, 0.12], *p* = 0.014). Then, for the duration of 1 to 2 h (B = 0.01, 95% CI [−0.06, 0.08], *p* = 0.750) as well as over two hours (B = 0.01, 95% CI [−0.04, 0.06], *p* = 0.766), it seemed to level off and was no longer significant. Thus, a moderate amount of physical activity, around 1 h per day, was associated with higher life satisfaction, but extending its duration further did neither increase nor decrease life satisfaction.

Third, we tested the effect of sleep with knots at 0, 7, 8, 9, and 12.5 h. Adding the effect of sleep duration significantly increased the explained variance, ΔR2 = 0.013, F(4, 2597) = 9.07, *p* < 0.001. The slope was slightly positive and significant for 0 to 7 h of sleep (B = 0.03, 95% CI [0.01, 0.05], *p* = 0.012), then increased for 7 to 8 h of sleep (B = 0.09, 95% CI [0.04, 0.15], *p* < 0.001), but decreased and became slightly negative and non-significant for 8 to 9 h (B = −0.02, 95% CI [−0.09, 0.04], *p* = 0.468) and 9 or more hours of sleep (B = −0.04, 95% CI [−0.09, 0.02], *p* = 0.238).

Fourth, we tested the effect of time spent in school with knots at 0.2, 5, 6, 7, and 11.5 h. However, adding the effect did not significantly increase the explained variance, ΔR2 < 0.001, F(4, 2571) = 0.35, *p* = 0.842. Finally, we tested the effect of school commuting with knots at 0, 0.5, 1, and 5 h. Again, adding this effect only led to a negligible and non-significant increase in explained variance, ΔR2 < 0.001, F(3, 2302) = 0.10, *p* = 0.960. Therefore, neither the time spent traveling from home to school and from school to home nor the time spent attending school seemed to affect life satisfaction.

Tables displaying all coefficients of the estimated models and their fit statistics can be found in the Appendix A.

## 4. Discussion

This paper aimed to examine cohort changes in the time spent on daily activities and their associations with life satisfaction, using four birth cohorts of Czech adolescents.

Our findings revealed the strongest birth cohort difference for sleep duration. The results clearly showed a decreasing trend in the average sleep duration per day between the cohorts. Specifically, in 1992, adolescents reported sleeping on average 8.6 h a day, while, in 2019, they reported sleeping on average 7.5 h a day, showing a difference of more than one hour. These results are consistent with the noted decline in sleep duration identified mainly among U.S. adolescents in the last decades [18,19,27]. Such findings may be interpreted in light of the increased use of new media and electronic devices that are easily carried into the bedroom and used before sleeping, resulting in delayed sleep time [59]. Sleep deprivation has become a significant public-health issue, since insufficient sleep duration may put adolescents at a greater risk for detrimental health and well-being outcomes [16,39,40]. Indeed, our results showed a significant positive association between sleep duration and life satisfaction. Specifically, adolescents’ life satisfaction increased up to the threshold of 8 h of sleep. After that, the association between life satisfaction and sleep duration became non-significant, suggesting adolescents aged 13–18 benefit most from 8 h of sleep. This observed sleep length is the minimum number of hours of sleep (i.e., 8 to 10 h of sleep per 24 h) that the American Academy of Sleep Medicine recommends for 13–17-year-old adolescents [41]. Interestingly, our study showed that 8 h or more of sleep duration did not affect adolescents’ life satisfaction. On the contrary, long sleep length has been associated with depressive symptoms [16], suggesting that a long sleep pattern in adolescence should not be overlooked, as it could indicate other mental health problems. However, besides sleep duration, sleep quality appeared to be even more critical in predicting the well-being of Czech adolescents [43]. In addition, other aspects such as intrapersonal factors (e.g., dispositional optimism), health-related factors (e.g., poor self-rated health, obesity, etc.), and parental and familial factors (such as family structure, parenting style, and parental emotional and social support) might play a role in the effect of sleep duration on life satisfaction [39,40,60].

Since concerns have been raised about adolescents’ time spent on screen-based activities, at the expense of the time dedicated to physical activities [36], our study specifically addressed leisure activities, focusing on sports and online gaming (i.e., actively and passively spent leisure time). Although cohort differences in time spent on sports activities were rather negligible in our study (η^2^ around 0.01), 54% of the participants did not report any sports activity during the day. This finding is significant for two reasons. First, the involvement in sports activities among Czech adolescents has not changed over the last 30 years, even though public-health researchers draw attention to promoting physical activities in adolescence, due to their beneficial effects on mental and physical health [21]. Second, our study showed that around 1 h of physical activity per day is associated with higher life satisfaction, suggesting that even moderate physical activity may benefit adolescents’ well-being. This finding aligns with the WHO global guidelines [61], which recommend at least an average of 60 min per day of physical activity for adolescents. It is worth mentioning that extending the duration of sports activities neither increases nor decreases life satisfaction. This result corroborates previous research [37], in which engaging in physical activities was linked to higher well-being, though only to a certain threshold.

A threshold was also observed concerning the time spent on online gaming. Our study showed that internet gaming had a negative effect on life satisfaction, but only after a 1 h threshold. This result is in line with previous findings that differentiated between the level of well-being in groups of light (<1 h a day) and heavy users (5+ h a day) [33]. Our findings add to the research evidence suggesting that our negative perception of the time adolescents spend playing online games should be adjusted, and we should refrain from pathologizing screen-based gaming [62]. In addition, it is worth mentioning that 65% of the Czech respondents reported no involvement with online games at all during the day. Concerning leisure activities such as online gaming and sports, it is essential to mention that time spent on sports or in front of a screen can often involve social interactions with friends (e.g., sports teams, exercise partners, or playing online games) and, thus, become a social activity [63,64,65]. However, our study did not examine leisure-time companionship, so this variable could be added in future studies to better understand the relationship between time use and life satisfaction.

In the non-leisure category, we examined time spent in school and commuting. Minor cohort differences were observed in time spent in school, with the highest school attendance in the 2019 cohort, supporting the previous belief that time spent in school is increasing [46]. In an earlier study [46], adolescents spent approximately 5 h a day in school on average, while our results showed 6 h. Cohort differences in school commuting were non-significant. Moreover, regarding the non-leisure categories, neither school commuting nor time spent in school affected life satisfaction. Regarding school attendance, factors other than time spent in school might affect life satisfaction. More concretely, life satisfaction previously has been associated with school-level factors, such as type of school, school size, teacher qualification [66], school stress [67], school engagement [6], school performance [68], class climate [69], and others. Our results on school commuting did not support previous studies of adults [47,48] or adolescents [15]. However, children who are accompanied to a school rated their journeys as happier compared to their unaccompanied counterparts [70], suggesting that more research is needed on independent mobility from the adolescents’ perspective.

Regarding life satisfaction, differences in birth cohorts were relatively negligible (η^2^ around 0.01). Our results did not support the existing research on the decrease in adolescents’ well-being [34,35], because the 1992 and the 2011 cohorts showed significantly lower life satisfaction than the 2019 cohort. Besides the time spent on daily activities, some demographic variables were related to lower life satisfaction. Specifically, girls and adolescents who perceived their parents to earn less than others reported lower life satisfaction. Previous studies have suggested that the variations in the reported gender differences in life satisfaction could be due to studies adopting diverse methods, such as geographical region and population type [71]. Generally, gender differences tend to be larger in European and high school samples [71]. In addition, low SES has already been connected to lower life satisfaction [72], and some studies have shown an association between SES and time use. Specifically, adolescents from families with lower SES sleep less [22], engage more in screen-time behavior, and are physically less active than children from higher social classes [23].

This study’s results and conclusions need to be considered in the light of limitations that inform directions for future research. First, the participants attended schools located in the region of South Moravia, which lies in the eastern part of the Czech Republic and, according to economic and educational indices, places in the middle of the country’s regions; thus, the sample’s representativeness is limited. Additionally, the number of participants in each cohort differed significantly, with the highest number enrolled in the 2019 cohort. The first three samples are much smaller, limiting the possibilities for more detailed models. Moreover, while participants in the first two cohorts completed the pen-and-paper questionnaires, participants in the third cohort participated online, and participants in the fourth cohort filled out either a pen-and-paper questionnaire or its online version. Although we found no systematic differences between these two versions in the studied variables, there were some slight differences between the pen-and-paper and online responses. For example, there were more missing or non-valid responses in the online-collected data, and the online responses were more likely to come from larger cities (22% vs. 15%). The choice of the mode of administration was mainly based on convenience—whether a school had a computer lab available for the survey. All these weaknesses limit the generalizability of the findings. In addition, the collected data were cross-sectional, so the causal links between daily activities and life satisfaction could not be studied. We must also account for the self-reported character of the data. Though anonymous, the self-reports are not free from social desirability and other biases, especially in school settings where there is a tendency to discuss one’s responses with classmates after the survey. It should also be noted that the analyses do not account for the fact that whole classrooms were sampled, making error terms dependent. This is due to missing information about classroom membership in the first two cohorts, and the minimal variance explained by classroom membership in the latter two cohorts (ICC’s < 0.03). Despite these limits, the present study is the first European study that included various birth cohorts of adolescents to examine cohort changes in time spent on daily activities and their associations with life satisfaction.

To improve the well-being of adolescents, interventions are needed to promote recommended hours of sleep, sports activities, and online gaming. Although, in the Czech Republic, we can find a few programs that focus on support of adolescents’ mental health, these programs usually focus on specific population groups, such as inclusion [73]. However, a universal educational program teaching mental-health literacy recently has been developed by the National Institute of Mental Health in the Czech Republic [74]. Under the guidance of trained teachers, the program identifies school as an ideal platform for a universal intervention to improve the mental health of children and adolescents. Over the course of 20 lessons, five main themes are taught, including one on mental health. Moreover, parents are provided with necessary information on mental health in adolescents and how to support it. Since the program is currently under evaluation before being available to the public, the results of our study might provide valuable information for the program’s working group to incorporate. We believe that our research will support the implementation of such a program in most schools, to create a space in which adolescents and their parents might be counseled and educated about the influence of time use on mental well-being.

## 5. Conclusions

Our results showed that time spent on daily activities affects adolescents’ life satisfaction. More specifically, sleeping 8 h a day and engaging in sports activities for about an hour a day contributed to higher life satisfaction. Spending more than 1 h a day gaming online led to lower life satisfaction in Czech adolescents. In addition, the analyses of the differences between birth cohorts revealed a decrease in sleep duration of more than 1 h, pointing to a generational change in adolescents’ sleeping habits and the need for public-health programs to address this issue.

## Figures and Tables

**Figure 1 ijerph-19-09422-f001:**
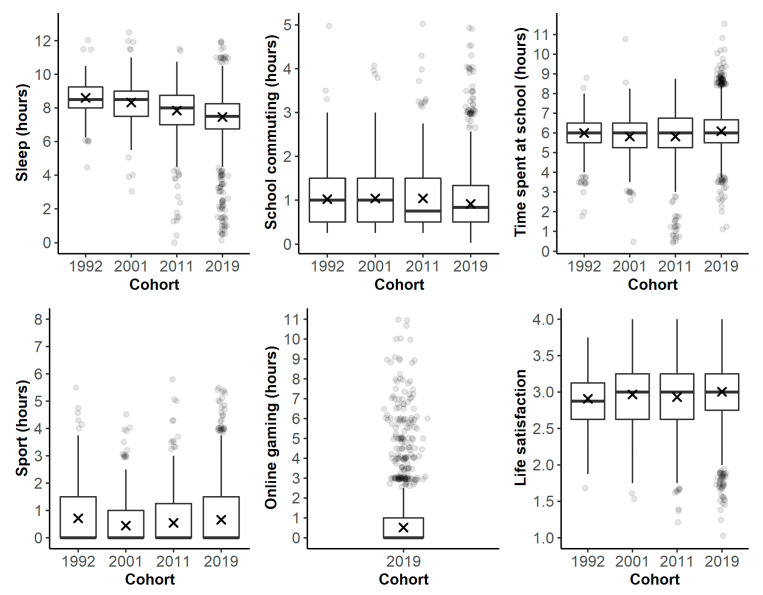
Duration of different daily activities and life satisfaction by cohort. Note. Online gaming was measured only in the last cohort (2019). The symbol “×” marks 5% trimmed means.

**Figure 2 ijerph-19-09422-f002:**
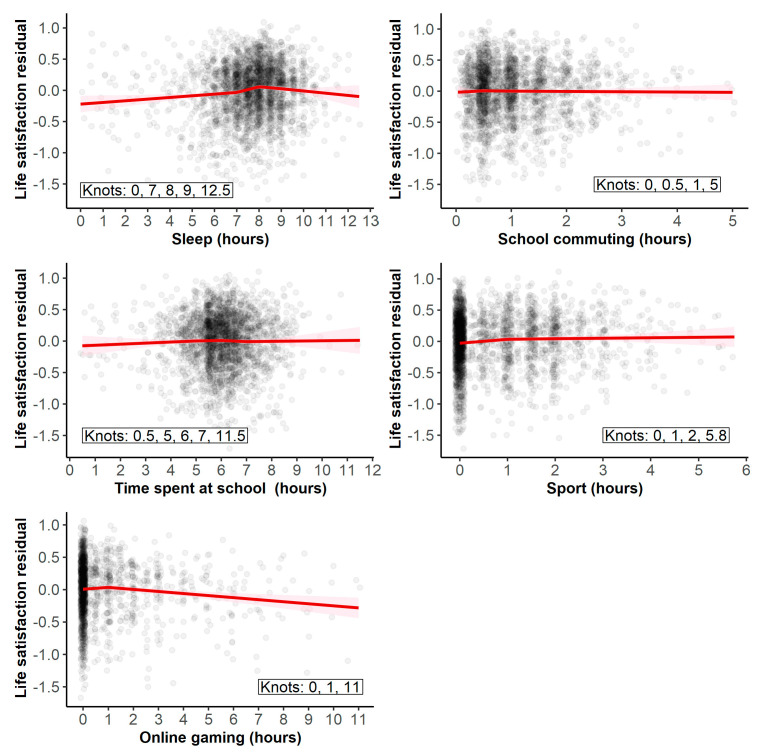
Effect of sleep, school commuting, time spent in school, sports, and gaming duration on life satisfaction. Note. The *Y*-axis shows the life satisfaction residuals, controlling for the effect of the cohort, gender, age, and parents’ income. The pink shaded regions indicate 95% confidence intervals for the fitted lines.

**Table 1 ijerph-19-09422-t001:** Sample characteristics.

Variable	Category	Cohort								Total	
		1992		2001		2011		2019			
		*n*	%	*n*	%	*n*	%	*n*	%	*n*	%
Gender	Boys	141	56	148	49	183	54	744	41	1216	45
	Girls	112	44	156	51	155	46	1058	58	1481	55
	No response	0	0	1	0.3	0	0	17	1	18	0.7
Age	13–15 years	99	39	129	42	79	23	466	26	773	28
	16–18 years	154	61	174	57	258	76	1345	74	1931	71
	No response	0	0	2	0.7	1	0	8	0	11	0.4
Parents’	Same as other parents	187	74	196	64	224	66	1217	67	1824	67
earnings	Less than other parents	39	15	53	17	57	17	221	12	370	14
	More than other parents	25	10	48	16	54	16	328	18	455	17
	No response	2	0.8	8	3	3	0.9	53	3	66	2.4

**Table 2 ijerph-19-09422-t002:** Comparison of cohorts using robust ANOVA and post hoc tests.

**Variables**	**Cohort**										
	**1992**					**2001**		**2011**		**2019**	
	**Mtr**				**SDw**	**Mtr**	**SDw**	**Mtr**	**SDw**	**Mtr**	**SDw**
Sleep	8.60				0.89	8.32	1.08	7.84	1.32	7.46	1.12
	[8.48, 8.72]					[8.19, 8.46]		[7.68, 8.00]		[7.40, 7.52]	
School commuting	1.02				0.64	1.04	0.66	1.04	0.76	0.91	0.63
	[0.93, 1.11]					[0.95, 1.12]		[0.94, 1.13]		[0.88, 0.95]	
Time spent at school	5.99				1.06	5.81	0.91	5.82	1.54	6.09	0.95
	[5.85, 6.14]					[5.70, 5.93]		[5.63, 6.01]		[6.04, 6.13]	
Sports	0.71				1.03	0.44	0.86	0.54	0.88	0.66	0.93
	[0.57, 0.85]					[0.34, 0.55]		[0.44, 0.65]		[0.61, 0.71]	
Online gaming										0,52	1.27
										[0.45, 0.58]	
Life satisfaction	2.91				0.33	2.97	0.39	2.93	0.42	3.00	0.41
	[2.86, 2.95]					[2.92, 3.02]		[2.88, 2.98]		[2.98, 3.02]	
**Variables**	**Robust ANOVA**				**Post Hoc Tests** **(Mean Differences with 95% Confidence Intervals Corrected for Multiple Testing)**						
	**F**	**dfM**	**dfR**	** *p* **	**η^2^**	**1992 vs. 2001**	**1992 vs. 2011**	**1992 vs. 2019**	**2001 vs. 2011**	**2001 vs. 2019**	**2011 vs. 2019**
Sleep	118.7	3	563.6	<0.001	0.159	0.27 *	0.76 ***	1.14 ***	0.48 ***	0.86 ***	0.38 ***
						[0.03, 0.52]	[0.49, 1.03]	[0.96, 1.32]	[0.20, 0.76]	[0.67, 1.06]	[0.15, 0.61]
School commuting	2.0	3	548.0	0.115	0.004	−0.02	−0.02	0.11	0.00	0.12	0.13
						[−0.18, 0.15]	[−0.19, 0.16]	[−0.02, 0.24]	[−0.17, 0.17]	[0.00, 0.25]	[−0.01, 0.26]
Time spent at school	14.2	3	521.9	<0.001	0.048	0.18	0.18	−0.09	0.00	−0.27 ***	−0.27 *
						[−0.07, 0.42]	[−0.14, 0.49]	[−0.30, 0.11]	[−0.30, 0.29]	[−0.44, −0.10]	[−0.53, −0.01]
Sports	5.3	3	553.4	0.001	0.012	0.27 *	0.17	0.05	−0.10	−0.22 **	−0.12
						[0.03, 0.50]	[−0.07, 0.40]	[−0.15, 0.25]	[−0.30, 0.10]	[−0.37, −0.06]	[−0.27, 0.04]
Online gaming											

Life satisfaction	6.1	3	565.9	<0.001	0.010	−0.06	−0.02	−0.10 **	0.04	−0.04	−0.07 *
						[−0.15, 0.03]	[−0.12, 0.07]	[−0.16, −0.03]	[−0.06, 0.13]	[−0.11, 0.03]	[−0.15, 0.00]

* *p* < 0.05, ** *p* < 0.01, *** *p* < 0.001.

## Data Availability

Data are available on reasonable request from the last author of the study (macek@fss.muni.cz).

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
