# Peer review of "Time Spent on Daily Activities and Its Association with Life Satisfaction among Czech Adolescents from 1992 to 2019"

_ijerph, 2022, doi:10.3390/ijerph19159422_

Round 1
Reviewer 1 Report
I commend the authors for conducting this study. The study has important implications for research and practice.
I have the following comments:
The Introduction could be structured better to establish links amongst the variables (e.g., life satisfaction, use of devices, sleeping habits, and participation in sport and exercise). As it stands now, the links seem a bit sporadic. Perhaps, focusing on the flow of the paper here would help. Please present the variables and their associations strategically and sequentially. Subsequently, please clarify how and why the three factors (lines 49-51) were chosen.
Perhaps include research (lines 52-66) from Czech adolescents too.
Clarify Internet gaming vs online gaming
Line 167: Write MLR in full the first time
Line 327: 'decades'
Lines 327-329: I wonder if the authors asked participants if they engaged in online gaming right before bedtime, and how that may have affected their sleeping habits.
Also, what other factors could have potentially affected/influenced the relationship between sleep and life satisfaction.
It would be great if the authors could outline, toward the end of the Discussion section, how this study would benefit future research and practice. More specifically, how can health researchers and policy developers use this information to design effective interventions and campaigns and inform public health initiatives.
Finally, please proofread the manuscript and cross-check the reference list.
Author Response
Reviewer 1 comments
- The Introduction could be structured better to establish links amongst the variables (e.g., life satisfaction, use of devices, sleeping habits, and participation in sport and exercise). As it stands now, the links seem a bit sporadic. Perhaps, focusing on the flow of the paper here would help. Please present the variables and their associations strategically and sequentially. Subsequently, please clarify how and why the three factors (lines 49-51) were chosen.
Author response: Thank you for your recommendation to improve the structure of the introduction. Following your recommendations, we rearranged the introduction so it flows more smoothly from one thought to another. We highlighted changes that were made (lines 27-133).
Regarding the clarification of chosen daily activities, we slightly changed the lines 131-133.
For the paper we selected only those activities that have been previously found to be associated with life satisfaction in adolescence. Although we could try to analyze all the activities there were several reasons not to do so. The primary reason is that with all the sampling issues it would be difficult to critically interpret any association that we could possibly find in activities for which we did not have sufficient theoretical background. It would force us to use p-value correction for multiple tests resulting in lower power. Still, the chance of both type I and type II errors would be higher. A secondary reason was to keep the manuscript brief and to the point.
- Perhaps include research (lines 52-66) from Czech adolescents too.
Author response: Thank you for this proposal. In the Introduction (lines 85-88), we now state how many of Czech adolescents meet sleep recommendations on school days. Moreover, we also added research from Czech adolescents in the paragraph on online gaming (lines 48-50).
- Clarify Internet gaming vs online gaming.
Author response: Thank you for this comment. In accordance with Lemmens and Hendriks (2016), we use these terms as synonyms. However, we now clarify it in the text (line 48).
References:
Lemmens, J. S., & Hendriks, S. J. (2016). Addictive online games: Examining the relationship between game genres and Internet gaming disorder. Cyberpsychology, Behavior, and Social Networking, 19(4), 270-276. https://doi.org/10.1089/cyber.2015.0415
- Line 167: Write MLR in full the first time.
Author response: Thank you for pointing out this deficiency. We now state Robust Maximum Likelihood (MLR) estimator (line 252-253).
- Line 327: 'decades'.
Author response: Thank you for pointing out this spelling mistake. We now state “decades” (line 413).
- Lines 327-329: I wonder if the authors asked participants if they engaged in online gaming right before bedtime, and how that may have affected their sleeping habits.
Author response: Thank you for this question. The effects of blue light have been well described in the literature and it would have been useful if we could observe them in our data. Since we have not asked our respondents about sleep quality the only data we have available in this respect is sleep length and the time elapsed between the end of the last reported gaming period and the beginning of sleep. For about 14% of respondents who reported at least one gaming period, the gaming was reported to end within one hour before the reported start of the sleep period. We found no systematic association between this and any sleep variable we had. Even if we did it would be very difficult to interpret it. The main reason would be that the reported „start“ time of the sleep period could be interpreted as the time of turning to bed by some respondents and as time of falling to sleep by others. Also, sleep length itself is only a very distant indicator of the quality of sleep or any problems with the circadian rhythm. In short, our data does not allow us to gain any insight into the effect of gaming on sleep other than the observation that a few respondents report long gaming period(s) followed by very short sleep before getting up for school, i.e. that it can be a cause of a lack of sleep.
- Also, what other factors could have potentially affected/influenced the relationship between sleep and life satisfaction.
Author response: Thank you for this comment. In a Discussion section, we now elaborate on other factors that could have play a role in the association between sleep and life satisfaction (lines 428-433).
- It would be great if the authors could outline, toward the end of the Discussion section, how this study would benefit future research and practice. More specifically, how can health researchers and policy developers use this information to design effective interventions and campaigns and inform public health initiatives.
Author response: Thank you for this idea. We have added a paragraph at the end of the Discussion section expanding on the public mental health programs in the Czech Republic that could benefit from our research (lines 523-538).
- Finally, please proofread the manuscript and cross-check the reference list.
Author response: Thank you for pointing out the insufficient language quality of the first version of the manuscript. Although the authors are not native speakers, we consider language quality to be very important. The revised version of the manuscript has been checked by a native speaker with extensive academic editing experience. In addition, we cross-checked the reference list and corrected mismatching references.
Reviewer 2 Report
The paper is very interesting and the datasets available are impressive even if the final samples are rather low for some of the waves. But some aspects have to be considered in order to improve literature review, methodological approach and the discussions.
Literature review is focused only on the importance of some daily activities for students’ wellbeing, and only a few mentions on the importance of socio-demographics. Some paragraphs on the importance of socio-demographics are needed also to better understand some of the findings.
There is a good matching in between the literature review and the model developed in the methodological part, and it is very well approached and developed. But some improvements could be made by extending the literature review and presenting for instance the importance of connectiveness with peers and time allocated to social and cultural activities. Data on dating and hanging around with friends are very valuable and at least according to other studies can explain variations in adolescents’ wellbeing. The information on online gaming has to be considered along with the information on time spend with friends in order to better understand and analyses the results on wellbeing variations.
Also if the datasets were used previously to analyse daily activities and wellbeing, some references on those papers must be added.
Also limits of the study must be mentioned, with a focus on the limits of the data and methodology used (volume of the sample, consistency of data, etc.).
There are some outputs of the data analyses included in the paper, but some others must be added for instance in an annex: how the index on wellbeing was constructed, estimation of the regressions, etc. Any output that can improve the reading and understanding of data analyses is recommended to be added in the annex. Also verify the Table 2 because seems to at least on mistake in the headings of the table.
Also in the discussion section would be very important to know if there were already implemented some public programs on adolescents mental health and wellbeing. Variations in wellbeing are low but it is important to understand if this is the case because some public interventions were carried out.
Author Response
Reviewer 2 comments
- Literature review is focused only on the importance of some daily activities for students’ wellbeing, and only a few mentions on the importance of socio-demographics. Some paragraphs on the importance of socio-demographics are needed also to better understand some of the findings.
Author response: Thank you for pointing out this deficiency. In line with your recommendation, we have added a paragraph on the role of socio-demographics in the relationship between time use and life satisfaction (lines 122-127). The information from this paragraph is further mentioned in the Discussion section to clarify some of the findings (lines 483-493).
- There is a good matching in between the literature review and the model developed in the methodological part, and it is very well approached and developed. But some improvements could be made by extending the literature review and presenting for instance the importance of connectiveness with peers and time allocated to social and cultural activities. Data on dating and hanging around with friends are very valuable and at least according to other studies can explain variations in adolescents’ wellbeing. The information on online gaming has to be considered along with the information on time spend with friends in order to better understand and analyses the results on wellbeing variations.
Author response: Thank you for this idea. Leisure companionship is indeed an important factor that needs to be considered in future studies on time use and life satisfaction. Especially engaging in screen time is often happening in the presence of friends, thus hanging out with friends and time spent online easily overlap. In accordance with your idea, we added this notion in the Discussion section to better understand the study results (lines 459-464). However, since we did not examine leisure companionship and we have tried to keep the manuscript brief and to the point, we did not include the literature review on spending time with peers.
- Also if the datasets were used previously to analyse daily activities and wellbeing, some references on those papers must be added.
Author response: Thank you for this comment. The datasets have not been previously used to analyze daily activities and well-being. However, some reports on the daily activities have been made using the very first dataset [references 12-13].
- Also limits of the study must be mentioned, with a focus on the limits of the data and methodology used (volume of the sample, consistency of data, etc.).
Author response: Thank you for alerting us to this issue. The paragraph at the end of Discussion has been focusing primarily on the limitations stemming from the non-probability sampling used in the surveys. We have now extended this paragraph to cover more limitations following from the sampling procedures and the self-report character of our data (lines 494-522).
- There are some outputs of the data analyses included in the paper, but some others must be added for instance in an annex: how the index on wellbeing was constructed, estimation of the regressions, etc. Any output that can improve the reading and understanding of data analyses is recommended to be added in the annex. Also verify the Table 2 because seems to at least on mistake in the headings of the table.
Author response: Thank you for this comment. We now elaborate on how the index of life satisfaction was constructed (lines 250-252). In addition, we now provide a Supplementary material containing 5 tables on linear regression with splines predicting life satisfaction based on time spent in daily activities (controlling for cohort, gender, age, and parents’ earnings).
Thank you for notifying us of the mistake in the Table 2 headings, it is now corrected. Moreover, the revised version of the manuscript has been proofread.
- Also in the discussion section would be very important to know if there were already implemented some public programs on adolescents mental health and wellbeing. Variations in wellbeing are low but it is important to understand if this is the case because some public interventions were carried out.
Author response: Thank you for this comment. However, we are not aware of any large-scale intervention in Czech schools that could have been affecting well-being over the past 30 years. Although the life-satisfaction SD's within cohorts appear low, it is mainy due to the 5% trimming used in the robust ANOVA. Without trimming, the SD's are around 0.5 across cohorts which is a common standard deviation on a 4-point scale from 1 to 4 when the distribution is negatively skewed. In published literature many different scales are used but generally the distribution is negatively skewed with the mean around 3/4 of the length of the scale and SD about 1/5 of the length of the scale.
Nevertheless, we have added a paragraph at the end of the Discussion section expanding on the public mental health programs in the Czech Republic that could benefit from our research (lines 523-538).
Reviewer 3 Report
The Authors of the paper submitted for review show how important a role for the life satisfaction in a group of adolescents is played by the way of meeting basic needs and the presence of healthy daily habits, also when controlling other demographic and psychological variables. The issues raised are extremely important, not only because of the searching for factors responsible for the life satisfaction of teenagers. The unique nature of the study - a long-term project - allows to track changes over time in the lives of adolescents, but it can also indirectly explain the increase in mental health problems of children and adolescents recorded in many countries in Europe and the world.
The article is well structured, synthetically, and precisely written. The authors are aware of some weaknesses of the project, which include different sizes of groups in subsequent cohorts and a large disproportion in the size of the last study group compared to those in previous years. I like that the Authors are therefore cautious in interpreting the results.
I formulated some doubts as questions:
1) In the description of the study, the authors indicate that the respondents could choose the form of the study in 2019. Were there differences in the results obtained with the paper-and-pencil method and with the online questionnaire? Were these results taken as an aggregate? There is no clarity in this regard.
2) The study groups seem to be homogeneous in terms of place of residence - a specific region in the Czech Republic. In addition, whole classes of students were examined. Was the selection of schools deliberate or random? Due to the study of whole classes - isn't it a situation that requires the use of statistics for dependent groups? Wouldn't it be worth emphasizing the characteristics of the region in which the research was carried out? Can its specificity affect the obtained results?
Table 2 in the second part of post-hoc tests shows a mistake - two columns are titled "1992 vs 2001".
Author Response
Reviewer 3 comments
1) In the description of the study, the authors indicate that the respondents could choose the form of the study in 2019. Were there differences in the results obtained with the paper-and-pencil method and with the online questionnaire? Were these results taken as an aggregate? There is no clarity in this regard.
Author response: Thank you for pointing this out. The description of the procedure was different and incorrect in the Discussion limitations paragraph, which caused confusion. We have corrected this and expanded a bit on the choice (lines 501-509). The choice was made mainly by the school. If the school did not have computer lab available or did not want to use it for the survey for whatever reason or preferred to administer the survey in a regular classroom, the pen-and-paper version was used. It was actually surprising that schools preferred the pen-and-paper mode. We checked for differences between the online and offline responses both in terms of individual variables and in terms of measurement models behind the summation scores. There were only minimal differences so they are only mentioned in the limitations not built into the models (esp. in the light of it being only a 2019 problem).
2) The study groups seem to be homogeneous in terms of place of residence - a specific region in the Czech Republic. In addition, whole classes of students were examined. Was the selection of schools deliberate or random? Due to the study of whole classes - isn't it a situation that requires the use of statistics for dependent groups? Wouldn't it be worth emphasizing the characteristics of the region in which the research was carried out? Can its specificity affect the obtained results?
Author response: Thank you for this comment. It is clear that sampling is the greatest limitation here. Although the intention was to achieve a representative sample of adolescents with respect to the type of school attended (at this age school attendance is compulsory) the non-probability quota sampling initially used in 1992 did not use a random selection. Schools were selected based on convenience as long as the proportion of vocational and grammar schools was the same as in population. The correspondence could only be rough since the number of participating schools was not high. Sampling is thus discussed as the primary limitation of our findings.
In relation to the question on the study the whole classes, we agree that sampling of whole classes should ideally reflect in the analysis because classmates/schoolmates tend to be more similar that randomly selected individuals from a population. There are two reason we did not apply multilevel models in our analysis. One is that for the first two cohorts the information about the class/school membership was not available. The other is the in the latter two cohorts where this information was available, the variance explained by school membership was very low, always bellow 5%. In situation we chose to keep the analysis simple. However, we should have mentioned this in the manuscript, which is now corrected (lines 515-519).
3) Table 2 in the second part of post-hoc tests shows a mistake - two columns are titled "1992 vs 2001".
Author response: Thank you for alerting us to this mistake, it is now corrected. In addition, the revised version of the manuscript has been proofread.